# Discrete Pseudohealthy Synthesis: Aortic Root Shape Typification and Type Classification with Pathological Prior

**Jannis Hagenah**                                                                    HAGENAH@ROB.UNI-LUEBECK.DE
**Floris Ernst**                                                                          ERNST@ROB.UNI-LUEBECK.DE
*Institute for Robotics and Cognitive Systems, University of Lübeck, Ratzeburger Allee 160, 23562 Lübeck, Germany*

## Abstract

In personalized prosthesis shaping, the desired shape remains typically unknown and has to be estimated based on the individual pathological shape. This estimation is also called pseudo healthy synthesis. One example application is the personalization of aortic root prostheses during valve-sparing aortic root surgery. Even though several methods for pseudohealthy synthesis were proposed during the last years, it might not always be necessary to taylor a completely individual and unique prosthesis for each and every patient as this introduces high costs and regulatory issues. Another option is to identify a set of prosthesis types that represents all natural healthy shapes in an adequate way. Then, the pseudohealthy synthesis problem becomes a classification problem, aiming on predicting the optimal prosthesis out of the set of candidates given a pathological shape. In this work, we present a fully automized workflow of unsupervised shape typification and type classification based on pathological data for the example of personalizing aortic root prostheses shapes. We provide a proof-of-concept study on an ex-vivo porcine data set, including a thorough evaluation of the model's hyperparameters and the number of identified shape types. Our study lies the groundwork for a new branch of personalized prosthesis shaping with a high potential of translation to clinical application: Discrete Pseudohealthy Synthesis.

**Keywords:** clustering, representation learning, personalized prosthetics, valve-sparing aortic root reconstruction

## 1. Introduction

The human body's morphology, including the morphology of its organs, varies strongly from patient to patient. All these individual organ shapes form a unique, fragile system that is optimized to work well together. This specifically holds for the human heart (Ni et al., 2018). Hence, if an organ or structure has to be replaced by a prosthesis, it is highly desirable that this prosthesis mimics the original morphology as close as possible to ensure optimal outcome for the patient. One example is the aortic root that shows an extremely high inter-patient variability (Scharfschwerdt et al., 2010). Together with the aortic valve leaflets that sit inside the root, a complex interplay ensures the prevention of blood flowing back into the ventricle and an optimal circulation in the whole body. Hence, a personalization of aortic root prostheses will help to keep this fragile system intact. However, the degree of personalization necessary to resemble the original organ shape sufficiently is unclear.

One typical issue in personalized prosthesis shaping is that the desired shape remains unknown as it can not be assessed using medical imaging. The only shape that can be acquired is the pathological state of the structure that should be replaced. As aortic root prostheses are mainly used for valve-sparing aortic root reconstruction (David and Feindel, 1992), the patient's aortic root is pathologically dilated and hence, manufacturing a prosthesis in this pathological shape does not help the patient at all. Even worse, the desired healthy shape remains unknown. Hence, it is necessary to estimate the individual healthy shape based on information of the pathological one, leading to a pseudohealthy synthesis problem.

The term pseudohealthy synthesis was introduced by (Bowles et al., 2016) and refers to the estimation of an expected healthy state based on pathological information. During the last years, several approaches were proposed to solve this problem (see 1.2). All these approaches have in common that the pseudohealthy state is estimated from a continuum, leading to a regression problem. However, tayloring a unique prosthesis for each and every patient poses enormous challenges on the translation to clinical application. Not only comes this completely individual tayloring with high costs and logistical efforts, but also does it raise regulatory questions. To tackle these problems, we propose to provide only a specific set of typical prosthesis shapes and estimate the optimal shape type for each patient, leading to a classification problem. As all of these prosthesis types could be manufactured in high numbers and could easily get regulatory certificates, such an approach is highly promising regarding the clinical applicability of personalized prosthesis shaping. Accordingly, the continuous pseudohealthy synthesis problem becomes a discrete classification problem to estimate the optimal prosthesis type based on pathological information.

In this work, we present a framework for fully automatic shape typification as well as type classification based on a pathological prior. We apply this method to the problem of personalized aortic root prosthesis shaping and provide a proof-of-concept-study, including an analysis of the hyperparameters.

## 1.1. Contribution of this Work

The contribution of this work is twofold. First, to the best of our knowledge, we present the first approach to formulate pseudohealthy synthesis in a discretized way aimed at prosthesis shaping. As the developed framework is not limited to aortis root prosthesis shaping, it could be applied to a wide range of organ shape synthesis problems from a pathological prior and hence provides a new method for pseudohealthy synthesis in general. Second, we see a high clinical value of our study for the development of personalized aortic root prostheses. The usage of a set of prosthesis types has a high potential for translation to clinical application and hence is of high interest for prosthesis manufacturers as well as clinicians.

## 1.2. Related Work

The term pseudohealthy synthesis was introduced by (Bowles et al., 2016) and refers to the synthesis of an image showing an estimated healthy state of an organ only based on surrogate information, which is typically given by the individual pathological state. Most approaches for pseudohealthy synthesis use a form of representation learning, either utilizing

autoencoders (Schlegl et al., 2017), (Baur et al., 2019), (Uzunova et al., 2019), (Chen and Konukoglu, 2018) or by training generative adversarial networks (GANs) (Andermatt et al., 2019), (Vorontsov et al., 2019), (Sun et al., 2020),(Xia et al., 2020). All these approaches have in common that they solve a regression problem, i.e. synthesizing the optimal image out of a continuous space. In contrast, we propose to only allow a finite set of discrete shapes, leading to a classification problem. To the best of our knowledge, the only study on pseudohealthy synthesis for aortic root prosthesis shaping also features a continuous prediction, solving a regression problem (Hagenah et al., 2019b).

The concept of shape typification by performing clustering in a latent space description was proposed in (Hagenah et al., 2019a). However, this has never been applied to aortic root shapes and, most of all, we present the first study to use this for type classification aimed at personalized prosthesis shaping.

## 2. Material and Methods

In this section, we present the developed framework in detail. After a description of the data set used in this study, we first focus on the automatic shape typification, followed by an explanation of the type classification from a pathological prior. Fig. 2 in appendix A shows an overview of our proposed framework.

### 2.1. Data Set

The data set used in this study was published in (Hagenah et al., 2016) and consists of ultrasound volumes of 24 ex-vivo porcine aortic roots. During data collection, the aortic root was extracted from the heart, attached inside a water bassin and put under physiologically realistic diastolic pressure. In this closed state, a 3D ultrasound image was acquired, mimicking a transesophageal echocardiography (TEE) examination. Afterwards, the root was artificially dilated by performing vertical cuts along the three sinus and sewing in diamond-shaped tissue patches, which is a common procedure for simulating aortic root aneurysms (Labrosse et al., 2015). After this dilation, another image was acquired in the same setup. Hence, for each of the 24 aortic root, the pathologically dilated as well as the healthy ground truth state is known.

In the scope of this study, only the horizontal slice image through the root that shows the commissure plane, i.e. the image plane that shows all three commissure points, is used. For these 2D images, a manual segmentation of the aortic root shape is available for the healthy roots, making a quantitative analysis more reliable. Previous studies showed the high information content of this slice (Hagenah et al., 2018). The images were resized to $96 \times 96$ pixels with a resolution of $1.27 \frac{mm}{pixel}$. The gray values were scaled to lie in the interval $[0, 1]$. We will denote the images as $I_i$ and the pathological ones as $\tilde{I}_i$ with $i = 1, \ldots, 24$, respectively. All images used in this study can be found in appendix D. We split the dataset into $N_{test} = 5$ aortic roots that served as a hold-out test set and 19 aortic roots for training and validation, referred to as training images.

## 2.2. Typification

The goal of the typification step is to identify a set of typical aortic root shapes that can serve as prostheses shapes. To automate this task, unsupervised clustering of the healthy roots could reveal these shape types. However, clustering directly in image space is insufficient as images are comparably high-dimensional and the clustering algorithm faces the curse of dimensionality (Assent, 2012). Thus, we follow the approach of clustering in a latent space description as proposed in (Hagenah et al., 2019a). Consequently, a latent representation of the healthy aortic root images is learned using a convolutional autoencoder. After encoding all healthy images into this low-dimensional representation, a clustering can be performed to identify typical shapes, i.e. the cluster centers, within the latent space. The desired shape of these potential prostheses can be synthesized using the decoder network. Thus, clustering can be performed in a meaningful representation while the similarity between a specific prosthesis and a real aortic root can be directly assessed in image space using typical metrics. The cluster centers, i.e. the prosthesis types, in latent space are denoted as $z_j^c$, $j = 1, \ldots, k$, where $k$ is the number of types. After decoding, the synthesized image of the $j$th shape type is called $\hat{I}_j^c$.

To this end, we propose a general, parameterized autoencoder architecture. The encoder $enc(I)$, $I \in \mathbb{R}^{96 \times 96}$ with $z_i = enc(I_i)$, consists of $n_c$ convolutional layers, each with $n_f$ filters, $ReLU$ activation and followed by a $2 \times 2$ average pooling. After a flattening operation, a dense layer follows where the number of neurons is the number of outputs of the last pooling layer. Then, a dense layer with $n_l$ and linear activation forms the bottleneck layer that outputs the latent representation $z \in \mathbb{R}^{n_d}$. The decoder $dec(z)$ with $\hat{I}_i = dec(z_i)$ follows the mirrored encoder architecture using upconvolution and upsampling, where $\hat{I}_i$ is the reconstructed healthy image. For training, we used the *adam* optimizer (Kingma and Ba, 2014), mean squared error loss, a batch size of 12 and we trained for 100 epochs.

To assess the influence of the hyperparameters $n_c$, $n_f$ and $n_l$, we evaluated several combinations of them regarding the performance of the resulting architecture on encoding and reconstructing unknown aortic root images. Thus, we performed a 10-fold Monte-Carlo crossvalidation on the training images (80% training, 20% validation) for each combination of hyperparameters examined, given in Table 1(a). We trained on the autoencoder on the training data, propagated the test data through the full network and computed the average root mean square error (RMSE) between the output and the original test images in each fold.

After identifying the optimal architecture, we examined the influence of the number of cluster centers, i.e. prosthesis types, on the capability of covering the full variance of observed root shapes by a small number of shape types. Following (Hagenah et al., 2019a), we utilized *k-means* clustering in the latent space. We evaluated different values of $k$ in a 10-fold Monte-Carlo crossvalidation (80% training, 20% validation) on the healthy training images. Within each fold, the autoencoder was trained on the training data using optimal hyperparameters, the training data was encoded to the latent space and *k-means* clustering was performed to identify $k$ shape types. Then, the $N_{test}$ test images were encoded and assigned to their respective clusters $z_j^c, j = 1, \ldots, k$. Finally, images of the prosthesis types $\hat{I}^c$ were synthesized by propagating the cluster centers through the decoder and each test image $I_i, i = 1, \ldots, N_{test}$ was compared to its corresponding prosthesis type image. We

Table 1: Examined hyperparameter values for the autoencoder (a) and the classification CNN (b). All combinations were assessed using 10-fold crossvalidation on the training images. The optimal combination regarding the RMSE between the original and the reconstructed image is marked in bold, respectively.

|  (a) | | | (b) | |
|---|---|---|---|---|
| **Parameter** | **Values** | | **Parameter** | **Values** |
| $n_c$ | $\mathbf{2}, 3, 4$ | | $m_b$ | $3, \mathbf{4}, 5$ |
| $n_f$ | $\mathbf{16}, 32$ | | $m_c$ | $\mathbf{1}, 2, 3$ |
| $n_l$ | $\mathbf{20}, 40, 60, 80, 100, 120, 140$ | | $m_f$ | $\mathbf{16}, 32$ |
| | | | $m_d$ | $1, 2, \mathbf{3}$ |
| | | | $m_n$ | $50, 100, \mathbf{150}$ |

performed this for $k = 1, \ldots, 20$ and used four different metric for the image comparison: The Jaccard Similarity, the Hausdorff Distance, the RMSE and the average symmetric contour distance (ASCD). To compute Jaccard Similarity, Hausdorff Distance and ASCD, the synthesized prosthesis image was segmented using thresholding ($t = 0.31$, corresponds to a grayscale value of 80) and compared to the segmentation of the test image, respectively. This holds for all calculations of these metrics within this work.

### 2.3. Type Classification

For discrete pseudohealthy synthesis, we propose to train a convolutional neural network (CNN) to predict the individually optimal prosthesis type based on an image of the pathological aortic root shape. As the decoder is known, it is possible to synthesize an image of the predicted prosthesis shape. In this study, we trained the CNN in a supervised way, where the ground truth label is retrieved by assigning each aortic root in the training set its corresponding cluster in latent space. Hence, in its current form, our method assumes that paired data is available, i.e. that for each patient in the training data set, the healthy as well as the pathological shape is known.

Once again, we propose a parameterized architecture. It follows a VGG-like structure (Simonyan and Zisserman, 2015) and consists of $m_b$ convolutional blocks, each consisting of $m_c$ convolutional layers with $m_f$ and *ReLU* activation followed by a $2 \times 2$ average pooling layer. After a flattening operation, $m_d$ fully-connected layers are following with $m_n$ neurons each and *ReLU* activation. Finally, the output layer is attached, featuring *softmax* activation. The training was performed using the *adam* optimizer, binary crossentropy loss and a batch size of 12.

We identified an optimal architecture by evaluating numerous combinations of the hyperparameters $m_b$, $m_c$, $m_f$, $m_d$ and $m_n$. Thus, as for the typification, we performed a 10-fold Monte-Carlo crossvalidation over all training images (80% training, 20% validation) for each combination of the hyperparameter values given in table 1(b). Within each fold, we trained the autoencoder on the training data using the optimal hyperparameters identified as described in section 2.2. We performed clustering as explained above with a fixed value

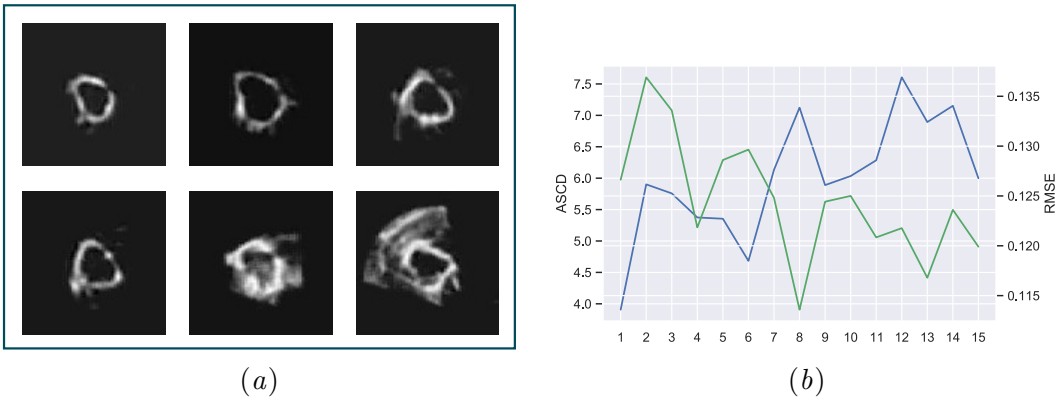

Figure 1: (a) Example of a set of identified shape types for $k = 6$. (b) Mean Similarity of healthy root shapes and their assigned prosthesis shapes for different values of $k$, given as the ASCD (blue, left axis) and the RMSE (green, right axis).

of $k = 6$ and assigned the optimal cluster centers to the training and the test images. Then, we trained the CNN on the training data to predict the optimal cluster center based on a pathological image. After training, we used the CNN to estimate the cluster centers for the pathological test data and assessed the classification accuracy. Additionally, we compared the healthy test images $I_i, i = 1, \ldots, N_{test}$ to the synthesized image of the predicted prosthesis $\hat{I}_j^c$ where $j$ is the classification result, respectively, once again using Jaccard Similarity, Hausdorff Distance, RMSE and ASCD as metrics.

It is important to note that due to the random data splitting during crossvalidation, the training dataset is typically imbalanced, i.e. some typical shapes and hence classes occur more often than others. To overcome this class imbalancing issue, we oversampled the minority classes in the training dataset so that all classes, i.e. prosthesis shapes, are represented equally (Gosain and Sardana, 2017).

## 3. Results and Discussion

As the data set used in this study contains paired data, a quantitative analysis is presented in addition to a qualitative analysis of the predicted and reconstructed images of the prosthesis shapes. The presentation of the results and their discussion is divided into typification and type classification, followed by an outlook focusing on practical challenges aiming on clinical application.

### 3.1. Typification

The hyperparameter analysis revealed that an architecture with $n_c = 2$, $n_f = 16$ and $n_l = 20$ provides the best image reconstruction accuracy with an RMSE of $0.09 \pm 0.03$. Fig. 1(a) exemplarily shows the synthesized images of a set of identified prosthesis shapes with $k = 3$. All of them look realistic and they complement each other well, indicating

that an automatic typification of aortic root shapes using clustering in latent space is possible. Given the optimal architecture of the autoencoder, Fig. $1(b)$ shows the capability of approximating all healthy test images by their respective prosthesis in dependency of the number of prosthesis shapes $k$. As expected, the RMSE decreases with increasing $k$. However, the ASCD increases, indicating that, regarding the shape, a smaller number of prosthesis types leads to better results. We identified $k = 6$ as an optimal value to assure reasonable values in both metrics. Average results for all metrics are given in Tab. 2. A visual assessment of the identified shape types is given as a t-SNE embedding in Appendix C.

### 3.2. Type Classification

Based on the optimal autoencoder architecture, the highest classification accuracy achieved during hyperparameter analysis for the CNN was $77.5 \pm 17.5\%$ with the parameters $m_b = 4$, $m_c = 1$, $m_f = 16$, $m_d = 3$ and $m_n = 150$. With a value of $67.1\%$, the classification accuracy was slightly lower on the holdout test dataset. Table 2 shows the similarity between the test images and the predicted prosthesis types for all four metrics. For comparison, the latter one is also given for the typification, i.e. with an optimal prosthesis choice. The classification accuracy indicates that the model is able to learn a relationship between the individual pathological root and the desired prosthesis shape. Additionally, the small difference between the classification model and the typification with optimally assigned prosthesis types shows that the classification works robustly and that most of the error relates to the discretization of the prosthesis shapes. Overall, the Jaccard-Similarity is low. This is most likely due to the relatively thin aortic root wall in the images. Even if a prosthesis approximates a root shape sufficiently, the overlap might still be quite small. Hence, the Hausdorff distance and the ASCD might be more meaningful on this data set. With an ASCD of around 4.5 pixels and a Hausdorff-Distance of around 8 pixels, the predicted prosthesis shapes seem to fit the desired root shapes adequately. Example prediction results can be found in appendix B. As a benchmark, we compared the discrete approach to a continuous, regression-based approach presented in (Hagenah et al., 2019b). Our proposed method outperforms the regression model by far. We assume that this is due to the implicit constraint to realistic and typical shape types. hence, no out-of-bag-predictions are possible in the discrete approach, making it more robust than the flexible regression approach.

### 3.3. Outlook

Overall, our results indicate that discrete pseudohealthy synthesis for personalized prosthesis shaping is possible. Specifically, the type classification seems to work adequately accurately, while most of the errors are introduced by the typification. This might be due to the very small dataset, as only 14 training samples are available within each fold. A bigger dataset might provide better typification results. Hence, data collection should be an important part of future work. Additionally, the transfer to human data is of high interest regarding clinical application.

Even though discrete pseudohealthy synthesis outperformed a regression approach on this dataset, the method is barely capable of dealing with out-of-bag samples, e.g. shapes

Table 2: Results of the comparison of healthy images and their corresponding prosthe­sis shapes for typification (optimal prosthesis) and type classification (estimated prosthesis) with $k = 6$, given as ASCD [$pixel$], Hausdorff Distance [$pixel$], Jaccard Similarity and RMSE. For comparison, the results are also given for the continuous approach proposed in (Hagenah et al., 2019b).

| Method | ASCD | Hausdorff | Jaccard | RMSE |
|---|---|---|---|---|
| Typification on validation set | $4.68 \pm 4.04$ | $14.48 \pm 8.62$ | $0.39 \pm 0.11$ | $0.13 \pm 0.05$ |
| Type Classification on test set | $4.48 \pm 2.59$ | $8.03 \pm 7.60$ | $0.43 \pm 0.09$ | $0.12 \pm 0.03$ |
| Continuous benchmark on test set | $24.39 \pm 39.07$ | $26.54 \pm 38.51$ | $0.17 \pm 0.11$ | $0.16 \pm 0.01$ |

that are very different from the identified typical ones. For these cases, a robust regression-based shape prediction might be a better choice.

As mentioned above, the classification method presented in this study assumes that the model can be trained on paired training data. With an increasing number of long-term screening studies with large cohorts, like for example the SHIP study (John et al., 2001), it is likely that paired data is available for a wide range of applications. However, extending our framework to be capable of dealing with unpaired data is an interesting research question and should be addressed in future work. Additionally, it might be possible to generalize from our study on the given small, paired dataset to larger, unpaired ones using transfer learning.

In this proof-of-concept study, the reliable error quantification was possible due to the available ground truth segmentation on the 2D image slices. Our method is easily extendable to also work on 3D volumes by only adding another dimension to all convolutional layers in the autoencoder as well as in the classification network. One might also think about utilizing geometric deep learning to process pointclouds instead of volumetric data. The framework stays the same, highlighting the flexibility of our approach.

## 4. Conclusion

In this work, we presented a novel approach for personalized prosthesis shaping, called discrete pseudohealthy synthesis. Thus, we proposed a framework for fully automatic shape typification and developed a type classification approach to estimate the optimal prosthesis type for a given pathological morphology. Furthermore, we provide a proof-of-concept study on personalized aortic root prosthesis shaping, including a vast hyperparameter analysis. Our results indicate that approximating the variance of natural aortic root shapes using a specific set of prosthesis types is possible and that the pathologically dilated aortic root shape carries enough information to classify the optimal prosthesis type only based on this dilated shape. As regulatory challenges and manufaction costs are way lower for a finite set of typical prostheses instead of fully personalized ones, our study presents an important step towards clinical application of personalized prosthetics.

## Acknowledgments

We want to thank Michael Scharfschwerdt for sharing his knowledge and expertise in the field of cardiovascular prosthetics. This work was supported by the KI-LAB Lübeck (funded by the German Federal Ministery of Education and Research (BMBF), grant number 01IS19069) by providing access to an NVIDIA DGX A100, where all computations presented in this work were executed.

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

## Appendix A. Sketch of Workflow

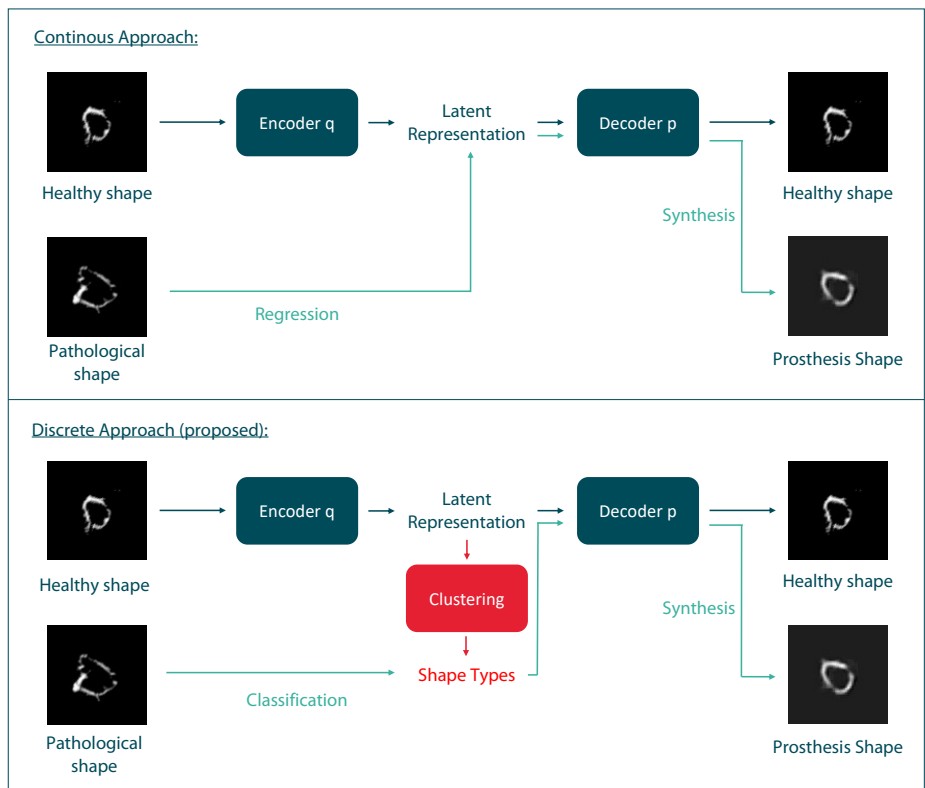

Figure 2: A sketch of the proposed method. The healthy images are encoded to a latent space representation. Previously published predict this latent representation continuously (top). In contrast, we propose to perform clustering in latent space to identify typical aortic root shapes (bottom). To compute the optimal prosthesis type for an individual patient, the shape type is classified based on an image of the pathological state and the corresponding image of the prosthesis can be synthesized.

## Appendix B. Qualitative Results

Fig. 3 shows qualitative results for two example folds.

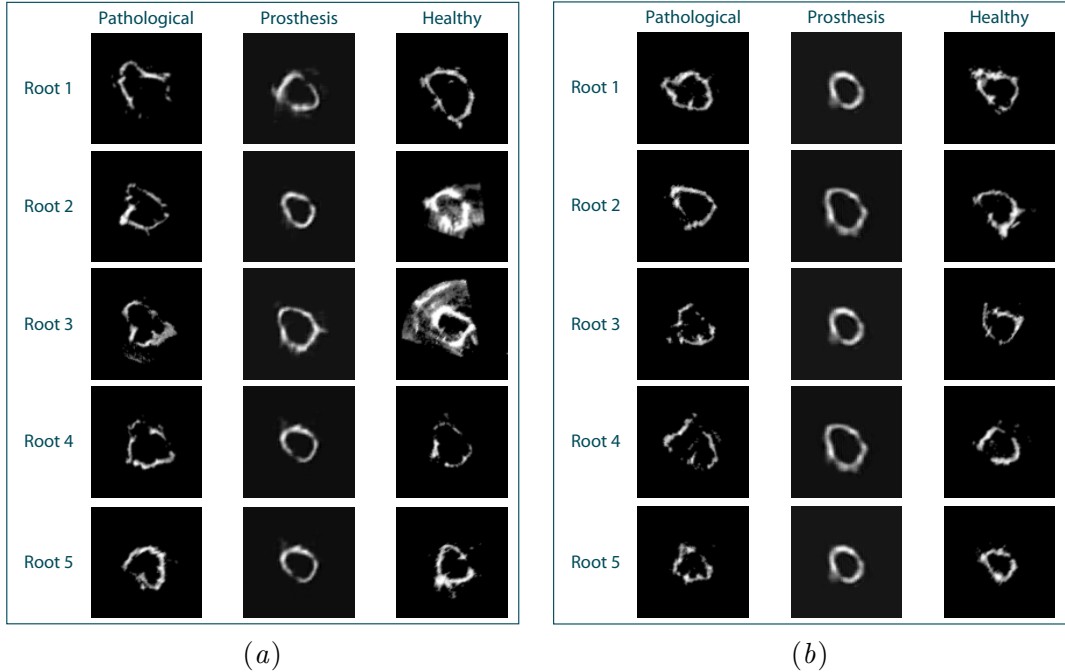

Figure 3: Qualitative results of type classification for the test images of two random folds (a) and (b). For each valve, the pathologically dilated shape, the predicted prosthesis shape as well as the healthy ground truth is shown.

## Appendix C. Cluster Assessment

Fig. **??** show the tSNE embedding of the healthy training images as well as their identified shape type, respectively, for $k = 6$. In general, the clusters are consistent and homogeneous. However, some shape types, e.g. cluster 1 or cluster 3, only appear very rarely, which might be due to the small dataset.

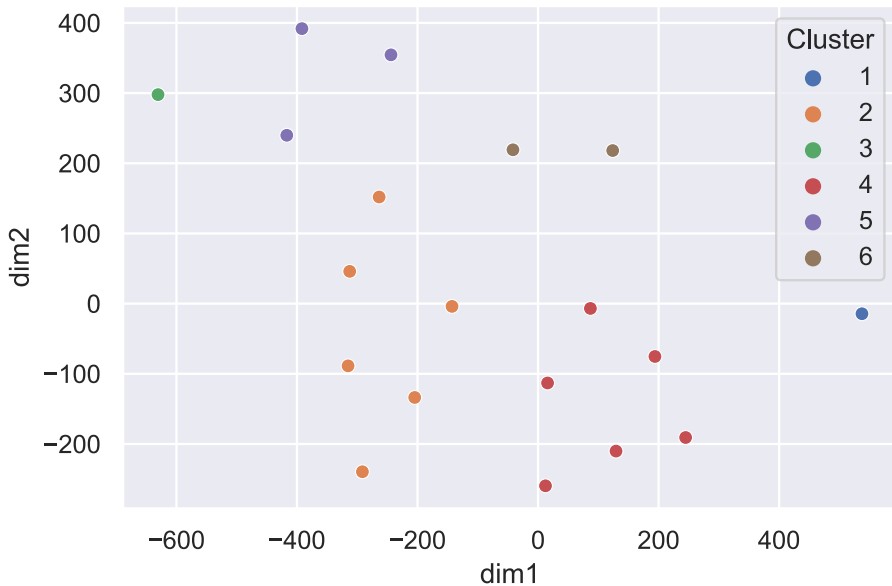

Figure 4: tSNE embedding of the healthy training images, separated by their respective cluster, i.e. shape type.

## Appendix D. Dataset

Fig. 5 shows the full dataset used in this study.

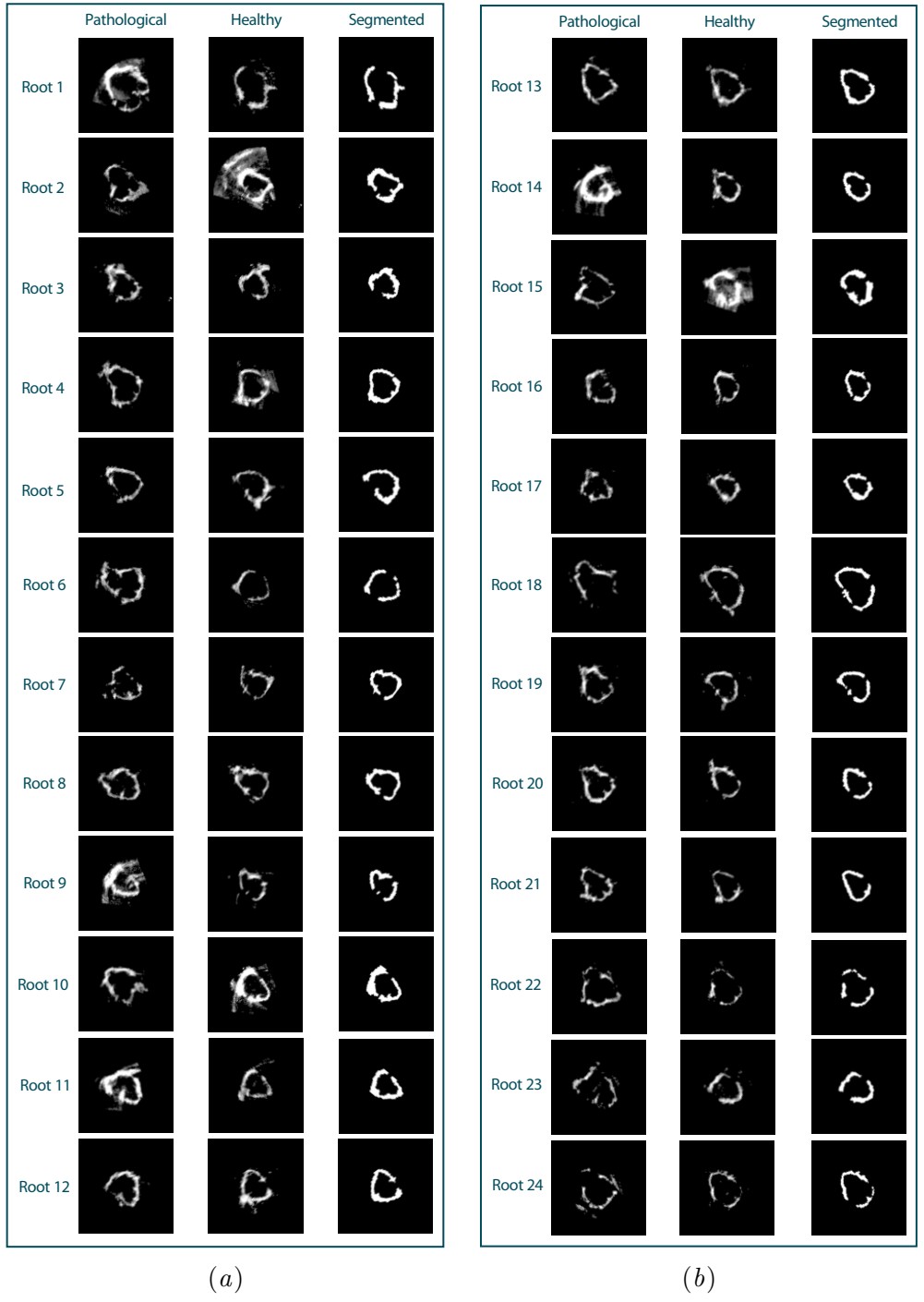

Figure 5: Full Dataset as it was used in this study.

