# OpenReview forum: "Discrete Pseudohealthy Synthesis: Aortic Root Shape Typification and Type Classification with Pathological Prior"
_MIDL.io/2021/Conference — MIDL 2021_

### Official Review · AnonReviewer3 · 2021-02-28

**Confidence:** 3
**Preliminary Rating:** 3
**Recommendation:** Oral

**Summary:**

This paper presents a novel method for prosthesis shaping applicable for any kind of organ shape synthesis problems. The proposed  framework is designed for a fully automatic shape typification and included a new classification approach to estimate the optimal prosthesis type for a given pathological morphology.

**Strengths:**

Methodology is well explained and good references. Extensive experiments show the great potential of such method. Well written paper in terms of explaining the main motivation behind this work and the proposed solution.

**Weaknesses:**

Some mathematical symbols are not well defined: #convAE, #latentdim. It is not commonly used in deep learning papers. The use of a single mathematical letter is highly required. It is not good to use a full word (healthy) inside a mathematical symbol such as the one denoting the reconstructed healthy image. The main figure is poorly designed: it does not show the main contribution of the work.

**Deanonymize Review:**

no

**Detailed Comments:**

A small figure showing the comparison between existing works and the proposed method can be added to explain better the advantage of solving the problem in that way.

**Justification Of The Preliminary Rating:**

The idea is novel and has a great impact if deployed in real-world clinical problems. However, from an academic perspective the full paper needs more improvement such as detailing the figure describing the architecture, using math symbols in a universal way like other paper used to present them.

**Paper Type:**

methodological development

**Questions To Address In The Rebuttal:**

Please improve the main figure to show the important subsections of your framework. It is very abstract figure so there is a high need to detail it very well.

**Special Issue:**

yes

---

> ### Author Response · Authors · 2021-03-18
> **Individual Rebuttal Letter to Reviewer 3**
>
> Dear Reviewer 3,
>
> Thank you for your helpful feedback and the interesting ideas.
>
> You are right that the mathematical notation was quite unconventional. We decided for this to increase the easy understanding. However, we agree that naming conventions in the scientific community are very different. Hence, we adapted the notation accordingly. We replaced all #parameter names by single letters and found alternatives for the full-word-descriptions. We would be happy to hear your opinion on our revised nomenclature.
>
> Furthermore, we really like your idea of replacing our graphical abstract figure by a sketch that directly shows the contribution of our work compared to previously published methods. Hence, we revised the figure accordingly. Please note that due to the strict page limit and added material during the rebuttal, the figure is part of the appendix now.
>
> Once again, thank you very much for your valuable suggestions.

---

### Official Review · AnonReviewer2 · 2021-02-28

**Confidence:** 4
**Preliminary Rating:** 3
**Recommendation:** Poster
**Final Rating:** 3

**Summary:**

The authors propose a new method for discreet Pseudohealthy Synthesis. They analyse the existing dataset and cluster the types of Aortic Root Shape and then perform Type Classification.  The training is done in a supervised capacity and the evaluation on a porcine dataset that has both healthy, pathological paired data.

**Strengths:**

The paper provides a nice and novel approach to a well studied problem.
The evaluation of the method is adequate and without any major logical gaps.
Paper is well organised and easy to follow , there were no major language issues

**Weaknesses:**

The paper's motivation, ie why perform the task via a discreet-classification method and not a regression one could be reiterated towards the end and been put into context with the final evaluation.

The paper is lacking in comparison with other methods. The authors identify a series of methods that are regression based but do not put their proposed solution into context in terms of performance.

There is little discussion about the limitations of the method. Any discreet method could potentially suffer from generilizability issues when facing a case that does not conform nicely to the prior identified types. An evaluation along these lines would increase the strength of the paper considerably.

**Deanonymize Review:**

no

**Final Rating Justification:**

I thank the authors for both the rebuttal and the changes to the manuscript. I am satisfied with the quality of this paper, hence, I maintain my position in favour of acceptance

**Justification Of The Preliminary Rating:**

The paper is both novel and interesting. Apart from some flaws in the writing of the paper and the lack of comparison to other techniques the community stands to benefit from this alternative view on an existing problem.

**Paper Type:**

validation/application paper

**Special Issue:**

no

---

> ### Author Response · Authors · 2021-03-18
> **Individual Rebuttal Letter to Reviewer 2**
>
> Dear Reviewer 2,
>
> Thank you very much for your helpful feedback. As mentioned in the general rebuttal, we agree that a comparison to a continuous approach is crucial. Hence, we added this comparison to the manuscript, thanks for pointing this out.
> Furthermore, we revised the discussion accordingly. We integrated the comparison to the continuous approach and hence included your requested reiteration of the benefits of our proposed method. The lack of capability of dealing with out-of-bag-samples indeed is an important limitation of our method. Thanks for bringing this up. We also addressed this in the discussion.
> Once again, thank you very much for your comments.

---

### Official Review · AnonReviewer1 · 2021-03-04

**Confidence:** 4
**Preliminary Rating:** 2
**Final Rating:** 3

**Summary:**

## Ideas

The paper presents an aortic root shape clustering and classification method using a convolutional autoencoder and convolutional neural network. The convolutional autoencoder is trained to reconstruct healthy aortic root images acquired with ultrasound, and testing is done by clustering the trained images, projecting the test images into the latent clusters and reconstructing the cluster's center. By clustering the images projected in the latent space, the authors are able to obtain classes to train a convolutional neural network to project pathological aortic root images in the latent space.  From the projection of the CNN to the "classes" and their corresponding respective cluster center, a healthy reconstruction can be computed by the decoder for the pathological images.

## Experiments

The authors use a dataset of 24 ex-vivo porcine aortic roots to perform their experiment.  They present results for the reconstruction of the healthy images, classification accuracy for the pathological images and results for the reconstruction of healthy images from their pathological counterparts.

**Strengths:**

The main ideas of the paper are well presented:
- Shape reconstruction is not an end in itself as healthy and pathological shapes differ
- Exact per-patient shape reconstruction is not necessarily desirable as taylor-made prosthetics can be costly
- Mass-producing a couple of types of prosthetics and finding the right type of prosthetic needed for each patient could allow for cheap treatment

The experiment used a good amount of metrics to inform on the results presented, and the architecture is well presented, as are the hyperparameters used for the experiment.

**Weaknesses:**

Unfortunately, the paper only presents one experiment on a very limited dataset. Such an application paper should and must be much more rigorous in its experimentation. Throughout the paper, the authors make several claims on the potential applications of the method on prosthetic development. Notably, in section 1.1

> we present the first approach to formulate pseudohealthy synthesis in a discretized way aimed at prosthesis shaping. The developed framework can be applied to a wide range of organ shape synthesis problems from a pathological prior and hence provides a new method for pseudohealthy synthesis in general.

However, the presented work only looks at the very limited case of aortic roots with a very limited dataset. Such claims of potential general applications should be backed up by experiments on more than one dataset.

Moreso, the presented work is not compared against any other methods aimed at a similar task. This is especially concerning since the presented work iterates on and is closely related to previous work by the same authors [1,2]. As such, it is confusing why the authors did not atleast compare the presented work to previous iterations of their own work, adapted as needed, and demonstrate how the improvements presented here are better than what was done before. While [1,2] reduce the novelty of the proposed work, lack of novelty alone should not pose a problem. However, lack of novelty should be compensated by clear improvements, which are lacking here as no comparison is provided. Even though the paper presents itself as distinct because it uses classification instead of regression (c.f end of section 1.2), this is not enough to justify the lack of comparison.

Also, the abstract mentions

> Besides a proof-of-concept study on an ex-vivo porcine data set, we provide a vast evaluation of hyperparameters, including the number of identified shape types.

This sentence puts very little emphasis on the experiment and much more on the evaluation of hyperparameters. This is confusing as hyperparameter search is expected and not really a contribution. Moreso, the actual hyperparameter search was lost in the text (sections 2.2 and 2.3, same as the method's components description). If emphasis was truly to be put on the thoroughness of the hyperparameter search, it should have been given its own subsection but mostly, it should have been accompanied with much more details: while table 1 offers the considered hyperparameters, it does not reflect how each impacted not only the RMSE but other reconstruction metrics.

Some doubts must also be raised concerning the experimentation protocol. Notably,

> Thus, we performed a 10-fold Monte-Carlo crossvalidation (80% training, 20% test) for each combination of hyperparameters examined, given in Table 1(a). We trained on the autoencoder on the training data, propagated the test data through the full network and computed the average root mean square error (RMSE) between the output and the original test images in each fold.

This implies that the hyperparameters were found by using the test images as a validation and that no images were held out to provide the final metrics reported. If this is the case, it is unacceptable and the experimentation protocol must be revised to keep the test set separate from the validation and training sets. If this is not the case, then the sentence (and similar ones throughout the text) should be reworded to avoid misleading the reader.

Finally, despite being well presented, the motivations behind turning this problem into a classifcation problem instead of a regression problem are not backed up by results. For example, the optimal $k$ was found to be 4, which is quite low. This implies that only 4 shapes are enough to represent the whole gamut of possible aortic roots. Could this be backed up by histological studies ? However, even the authors themselves might suggest that classification poses a problem as section 3.2 includes

> Additionally, the small difference between the classification model and the typification with optimally assigned prosthesis types shows that the classification works robustly and that most of the error relates to the discretization of the prosthesis shapes.

This sentence attributes the error to the discretization of the prosthesis shape, which is the core idea of the proposed work. It seems the proposed work could be improved in several ways by not being a classification problem. Moreso, no work is done to assess the coherence of the clusters: the autoencoder does not include a term in its loss to promote tight and distinct clusters, nor is any assessment of the cluster shapes (with a t-SNE projection, for example) provided.

All in all, as mentioned previously, what the proposed work lacks in novelty it does not make up in experimentation or analysis.

[1]: J. Hagenah, M. Mehdi and F. Ernst, "Generating Healthy Aortic Root Geometries From Ultrasound Images of the Individual Pathological Morphology Using Deep Convolutional Autoencoders," 2019 Computing in Cardiology (CinC), Singapore, 2019, pp. Page 1-Page 4, doi: 10.23919/CinC49843.2019.9005819.

[2]: Hagenah, J., Kühl, K., Scharfschwerdt, M. & Ernst, F.. (2019). Cluster Analysis in Latent Space: Identifying Personalized Aortic Valve Prosthesis Shapes using Deep Representations. Proceedings of The 2nd International Conference on Medical Imaging with Deep Learning, in PMLR 102:236-249

**Deanonymize Review:**

no

**Detailed Comments:**

The text could be re-arranged and some parts omitted to improve readability. Notably

> All these individual organ shapes form a unique, fragile system that is optimized to work well together. This specifically holds for the human heart (Ni et al., 2018). Hence, if an organ or structure has to be replaced by a prosthesis, it is highly desirable that this prosthesis mimics the original morphology as close as possible to ensure optimal outcome for the patient

This is in contradiction with the core idea of the proposed work, which is discretizing the prosthetic shape.

> We apply this method to the problem of personalized aortic root prosthesis shaping and provide a proof-of-concept-study, including an extensive analysis of the hyperparameters.

As mentioned before, the term "extensive" should be toned down.

> All these approaches have in common that the pseudohealthy state is estimated from a continuum, leading to a regression problem. However, tayloring a unique prosthesis for each and every patient poses enormous challenges on the translation to clinical application. Not only comes this completely individual tayloring with high costs and logistical efforts, but also does it raise regulatory questions. On the other hand, it might already be enough to provide a specific set of typical prosthesis shapes and estimate the optimal shape type for each patient, leading to a classification problem.

This part is cryptic and could use some rewording

> The developed framework can be applied to a wide range of organ shape synthesis problems from a pathological prior and hence provides a new method for pseudohealthy synthesis in general. Second, we see a high clinical value of our study for the development of personalized aortic root prostheses. The usage of a set of prosthesis types has a high potential for translation to clinical application and hence is of high interest for prosthesis manufacturers as well as clinicians.

The streak of citations in section 1.2 should be formatted differently to avoid double parentheses.

> However, one might also think about different approaches for training the CNN without the need for paired data, e.g. using reinforcement learning.

This claim is out of place and should either be backed up or be removed.

> Within each fold, we trained the autoencoder on the training data using the optimal hyperparameters identified as described in section 2.2.

Why was the autoencoder retrained if only the CNN was being optimized ?

> Then, we trained the CNN on the training data to predict the optimal cluster center based on a pathological image. After training, we used the CNN to estimate the cluster centers for the pathological test data and assessed the classification accuracy.

Was the CNN trained to predict the class or the cluster center directly ? There is some confusion in the text that must be cleared.

> The hyperparameter analysis revealed that an architecture with #convAE = 3, #filtersAE = 16 and #latentdim = 80

This is repeated information from table 1

> provides the best image reconstruction accuracy with an RMSE of 0.07 ± 0.04.

Where does the RMSE of 0.07 come from ? It is not included in Table 2. Also, table 2 should be referred to in section 3.1 as it seemingly includes results for typification.

> Fig. 2(a) exemplarily shows the synthesized images

Why not display the cluster centers found with $k=4$, as it is the optimal $k$ parameter found ? Fig 2.(b) should be included in the section relating to the hyperparameter search.

Overall, the text doesn't seem as concise as it should be given the limited number of pages available. The sections should be better organized to delineate what pertains to clustering, image reconstruction or classfication, what belongs to experiments and what belongs to the hyperparameter search, etc.

**Final Rating Justification:**

The rebuttal and changes made to the paper have greatly improved the quality of the proposed work.

**Justification Of The Preliminary Rating:**

The overall quality of the paper is very weak. Very little improvement is brought up considering previous work, and not enough experiments back up what improvements there is. The text needs to be reworked and made more concise and clearer.

**Paper Type:**

validation/application paper

**Questions To Address In The Rebuttal:**

The points mentioned above, notably the lack of experiments and datasets, as well as the possible usage of testing data to find the optimal hyperparameters should be addressed. Moreso, the text should be reworked to address the minor comments above.

**Special Issue:**

no

---

> ### Author Response · Authors · 2021-03-18
> **Individual Rebuttal Letter to Reviewer 1**
>
> Dear Reviewer 1,
>
> Thank you very much for your thorough review, the critical and detailed comments and the helpful feedback.
> We agree that the small dataset size is a limitation of our study and it would be great to evaluate the proposed method on a broader range of datasets. However, we are not aware of a publicly available dataset for shape estimation with known pairs of healthy and pathological samples. If you know such a dataset, we would be happy for suggestions. But we agree that our claim from 1.1 is not supported by results from this paper. We reformulated this.
>
> As mentioned in the general rebuttal, we addressed both of your main points. Thank you for pointing out the lack of a holdout evaluation. Our aim was to maximize the number of samples in the training as well as the test set. We never evaluated a model on data it was trained on, and through the randomization of the Monte-Carlo crossvalidations, we hoped to keep the entanglement as small as possible. However, you are right that there are specific entanglements as the hyperparameters were optimized to fit the full dataset, including the test samples. Hence, we recalculated all results, performed the hyperparameter analysis using validation data and reported the performance of the full model on a holdout test set.
>
> Furthermore, we completely agree that a comparison to previously published, continuous prediction approaches is necessary to interpret the results in a meaningful way. As mentioned in the general rebuttal, we added this comparison and could show that our proposed method outperforms the regression approach by far. Thank you for raising this issue.
>
> This leads us to your comment regarding the errors our core idea introduces. In general, we always expect errors when we discretize continuous values, and organ shapes will not be an exception. Hence, assuming a perfect and optimal regression model, our proposed method will always provide lower accuracies in terms of the examined metrics, and this is what we want to address in the paper. However, our results show the superiority of the discrete approach. We assume that this is due to the implicit constraint that our model can only predict realistic, typical shapes while the regression might produce highly unrealistic roots. Hence, we show that even though an optimal regression should be more accurate in theory, the robustness of our method makes the discrete approach better, at least on this dataset.
> Furthermore, the image-based accuracy metrics are not the only features that make the methods comparable. As mentioned in the introduction, the discrete approach solves many issues that are relevant in clinical application, e.g. the costs and regulation.
>
> You mentioned that we did not integrate auxiliary constraints into the loss function of the autoencoder to provide better clustering results. This is definitely an interesting idea and leads to deep clustering approaches. In this study, we relied on the adequate performance of k-means clustering in the latent space of a standard autoencoder shown in (Hagenah et al. 2019a). However, evaluating different concepts for deep clustering will definitely part of our future work.
>
> We like your suggestion of assessing the cluster coherence. Hence, we added a plot of the t-SNE embedding of the latent space, including the samples and identified cluster centers, in the appendix exemplarily.
>
> We agree that the weighting in our contribution statement within the abstract was wrong, and the proof-of-concept study should be the clear focus instead of the hyperparameter analysis. We rephrased this, thanks for mentioning.
>
> We appreciate all your detailed comments on wording and phrasing. We gave our best to address these points in the revised manuscript.
>
> Once again, thank you very much for your detailed and critical comments. Our manuscript highly benefits from your suggestions.

---

> > ### Comment · AnonReviewer1 · 2021-03-20
> > **Missing pieces**
> >
> > Thank you for addressing the comments above. The changes made according to the reviewers' comments have indeed improved significantly the readability and overall quality of the presented work. However, Table 2 now seems to omit the Type Classification results, even though its caption and the text refers to the classification accuracy. Moreover, the appendices seem to omit the t-SNE embedding visualization mentioned in the author's comment.

---

> > > ### Author Response · Authors · 2021-03-23
> > > **Sorry for mistake, thanks for detailed review**
> > >
> > > Thank you very much for your highly detailed and thorough review - again! We are sorry that there were these versioning errors and the inconvenience for you!
> > > We just uploaded the right version of the manuscript.
> > >
> > > We excluded the classification accuracy from Table 2 for optical reasons. In the new version, it is also removed from the caption and given in the text of section 3.2.
> > >
> > > The new version also shows the t-SNE embedding of the shape typification in the appendix.
> > >
> > > So, once again: Sorry and thanks!

---

### Official Review · ~Namkug_Kim1 · 2021-03-08

**Confidence:** 4
**Preliminary Rating:** 3
**Recommendation:** Oral

**Summary:**

The aim of this study is  to present a fully automized workflow of unsupervised shape typification and type classification based on pathological data for the example of personalizing aortic root prostheses shapes. Idea is very good.  The authors present to formulate pseudohealthy synthesis in a discretized way aimed at prosthesis shaping.
Second, the development of personalized aortic root prostheses could be used. The usage of a set of prosthesis types has a high potential for translation to clinical application and hence is of high interest for prosthesis manufacturers as well as clinicians.

**Strengths:**

The authors present to formulate pseudohealthy synthesis in a discretized way aimed at prosthesis shaping.
Second, the development of personalized aortic root prostheses could be used. The usage of a set of prosthesis types has a high potential for translation to clinical application and hence is of high interest for prosthesis manufacturers as well as clinicians.

**Weaknesses:**

Relatively small number of aorta roots including ultrasound volumes of 24.
There is a lack of ablation study, evaluation of parameter changes, and no external validation.
In addition, there is a lack of evaluation method on technical and clinical purposes.

**Deanonymize Review:**

yes

**Detailed Comments:**

The aim of this study is  to present a fully automized workflow of unsupervised shape typification and type classification based on pathological data for the example of personalizing aortic root prostheses shapes. Idea is very good.
The authors present to formulate pseudohealthy synthesis in a discretized way aimed at prosthesis shaping.
Second, the development of personalized aortic root prostheses could be used. The usage of a set of prosthesis types has a high potential for translation to clinical application and hence is of high interest for prosthesis manufacturers as well as clinicians.
Relatively small number of aorta roots including ultrasound volumes of 24.
There is a lack of ablation study, evaluation of parameter changes, and no external validation.
In addition, there is a lack of evaluation method on technical and clinical purposes.

**Justification Of The Preliminary Rating:**

The aim of this study is  to present a fully automized workflow of unsupervised shape typification and type classification based on pathological data for the example of personalizing aortic root prostheses shapes. Idea is very good.
The authors present to formulate pseudohealthy synthesis in a discretized way aimed at prosthesis shaping.
Second, the development of personalized aortic root prostheses could be used. The usage of a set of prosthesis types has a high potential for translation to clinical application and hence is of high interest for prosthesis manufacturers as well as clinicians.
Relatively small number of aorta roots including ultrasound volumes of 24.
There is a lack of ablation study, evaluation of parameter changes, and no external validation.
In addition, there is a lack of evaluation method on technical and clinical purposes.

**Paper Type:**

both

**Questions To Address In The Rebuttal:**

Relatively small number of aorta roots including ultrasound volumes of 24.
There is a lack of ablation study, evaluation of parameter changes, and no external validation.
In addition, there is a lack of evaluation method on technical and clinical purposes.

**Special Issue:**

no

---

> ### Author Response · Authors · 2021-03-18
> **Individual Rebuttal Letter to Reviewer 4**
>
> Dear Reviewer 4,
>
> Thank you for your comments. You are right that the dataset used in this study is quite small and an evaluation on a large dataset would be beneficial. However, we are not aware of another publicly available dataset for organ shape estimation with paired healthy and pathological data. If you know such a dataset, we are very happy for your suggestions.
> As mentioned in the general rebuttal, we recalculated our results on a holdout test set to address your request for an external validation. Additionally, we tackled the evaluation of parameter changes via the thorough hyperparameter analysis.
> Unfortunately, we are not completely sure what you mean by an “evaluation method on technical and clinical purposes”. Could you please clarify so that we can address this comment?
> Thank you very much.

---

### Author Response · Authors · 2021-03-18
**General Rebuttal Letter to all Reviewers**

Dear Reviewers,

first of all, thank you very much for your thorough review and the valuable and helpful feedback. We revised our manuscript according to your comments and believe that it highly benefits from your advices.
Throughout all reviews, we identified two main points of criticism that we want to address in this general statement.

1) You mentioned that our method was not compared to a previously published, regression-based approach for a better embedding of our study into the state-of-the-art. We completely agree that this comparison is highly relevant for interpreting the results in an adequate way. Hence, we computed the accuracy for the continuous approach presented in (Hagenah et al., 2019b). We could show that the discrete approach outperformed the regression-based approach by far. The results are given in Tab. 2 and we comment on them in the results and discussion sections.

2) You questioned the results as the dataset was very limited and no external validation on a hold-out test set was done. We agree that this was a limitation of our study. Hence, we recalculated our results while excluding a holdout test set from the data. The hyperparameter analysis was performed on the training images only, utilizing a crossvalidation to get multiple train/validation splits. The final results, i.e. the accuracy of our proposed method, was assessed on the holdout test set.

Besides these general comments on major revisions, we addressed your other points in individual rebuttals that you can find below.

---

### Meta-Review · Area_Chairs · 2021-03-29

**Recommendation:** Accept (Poster)

**Metareview:**

The paper receives unanimously positive comments from four experts, who are both knowledgeable and independent. It presents an unsupervised shape typification and type classification method for personalized prosthesis shaping. The reviewers hail at the potential of the proposed approach for clinical translation. Earlier there was a major concern regarding the limited evaluation. But, the authors provided an updated version to improve the quality per reviewing comments, which appease the reviewers' concerns.
Hence, I recommend its acceptance.


some typos:
taylor -> tailor
lies the groundwork -> lays

**Paper Type:**

both

---

### Decision · Program_Chairs · 2021-03-31

Accept